# Scalable Exploration for High-Dimensional Continuous Control via Value-Guided Flow

**Yunyue Wei**  **Chenhui Zuo**  **Yanan Sui**

Tsinghua University

yunyuewei@mail.tsinghua.edu.cn
zuoch22@mails.tsinghua.edu.cn
ysui@tsinghua.edu.cn

## Abstract

Controlling high-dimensional biological and robotic systems is challenging due to expansive state–action spaces, where effective exploration is critical. Commonly used exploration strategies in reinforcement learning are largely undirected with sharp degradation as action dimensionality grows. Many existing methods resort to dimensionality reduction, which constrains policy expressiveness and forfeits system flexibility. We introduce Q-guided Flow Exploration (QFLEX), a scalable reinforcement learning method that conducts exploration directly in the native high-dimensional action space. During training, QFLEX traverses actions from a learnable source distribution along a probability flow induced by the learned value function, aligning exploration with task-relevant gradients rather than isotropic noise. Our proposed method substantially outperforms representative online reinforcement learning baselines across diverse high-dimensional continuous-control benchmarks. QFLEX also successfully controls a whole-body human musculoskeletal model to perform agile, complex movements, demonstrating superior scalability and sample efficiency in very high-dimensional settings. Our results indicate that value-guided flows offer a principled and practical route to exploration at scale.

## 1 Introduction

Controlling over high-dimensional dynamical systems underpins a broad range of applications in robotics, sports, and embodied intelligence from legged locomotion to full-body musculoskeletal control. Complex sensorimotor coordination and over-actuation are common in such systems. As the number of sensors and actuators grows, these systems gain flexibility and robustness, enabling agile and precise movements. However, increasing dimensionality also amplifies the challenges of coordination and learning efficiency due to rapidly expanding state–action spaces, making effective exploration strategies essential.

A widespread practice in online deep reinforcement learning (RL) is to inject *undirected* stochasticity (e.g., Gaussian noise) into policy outputs for exploration (Haarnoja et al., 2018). While simple and effective in moderate dimensions, such isotropic perturbations rapidly lose coverage and become sample-inefficient as action dimensionality and actuator redundancy grow, yielding vanishing signal for discovering task-relevant actions. Dimensionality reduction-based learning constrains control within low-rank subspaces to make search tractable (Berg et al., 2024). Such complementary strategy may forfeit the flexibility and redundancy that high-dimensional dynamical systems are designed to provide.

Iterated sampling techniques, most notably diffusion models and flow-based transports, have achieved striking success in high-dimensional generative modeling, providing robust procedures for sampling in thousands of dimensions (Song et al., 2021; Lipman et al., 2023). Motivated by these advances, several works have adapted iterated-sampling ideas to control and decision making (Janner et al., 2022; Yang et al., 2023). Despite promising results in moderate-dimensional set-

tings, these methods have not demonstrated success for high-dimensional continuous control with substantial over-actuation.

In this paper, we introduce Q-guided Flow Exploration (QFLEX), a scalable exploration mechanism that operates in the *native* high-dimensional action space. QFLEX achieves directed exploration for policy improvement by sampling from probability flow induced by learned state–action value function $Q$, proposing informative actions aligned with task-relevant direction. Our design preserves flexibility of the complex systems, and integrates cleanly into a actor-critic loop, yielding efficient learning across diverse high-dimensional continuous-control benchmarks. QFLEX also succeeds in controlling a full-body musculoskeletal system to perform complex movements, highlighting its scalable and efficient exploration.

**Our contributions**: (1) We propose QFLEX, a scalable RL method which achieves value-aligned directed exploration with policy-improvement validity, enabling efficient learning over high-dimensional state-action spaces. (2) We present an actor–critic implementation of QFLEX that consistently outperforms representative Gaussian-based and diffusion-based RL baselines on a wide range of high-dimensional continuous-control benchmarks. (3) We demonstrate QFLEX on a full-body human musculoskeletal model with 700 actuators, achieving agile, complex movements and efficient exploration without dimension reduction.

## 2 RELATED WORK

**High-dimensional over-actuated control.** The control of high-dimensional dynamical system is challenging due to its high-dimensionality and over-actuation. With few model-based strategies (Hansen et al., 2024; Wei et al., 2025), model-free deep reinforcement learning is the mainstream solution for solving complex control tasks (Kidziński et al., 2018; Geiβ et al., 2024; Caggiano et al., 2024). Hierarchical RL decomposes decision making into high-level planning and low-level control, reducing exploration burden by restricting search to joint- or skill-level choices (Lee et al., 2019; Park et al., 2022; Feng et al., 2023). Curriculum-based learning iterates over sub-tasks to smooth the learning curve for diverse skill learning over high-dimensional embodiment (Caggiano et al., 2023; Park et al., 2025). DEP-RL employs bio-inspired sampling for coordinated exploration (Schumacher et al., 2023b;a). Lattice generated correlated noise for exploration by injecting stochasticity into latent embeddings of the policy network (Chiappa et al., 2023b; Simos et al., 2025). Synergy-based approaches such as DynSyn (He et al., 2024) learn or impose low-dimensional control subspaces derived from morphology or task structure, enabling more stable training on systems with high degrees of freedom. These methods primarily mitigate undirected exploration issue by explicit or implicit dimensionality reduction, which can constrain policy expressiveness and underutilize redundancy—potentially limiting the flexibility required for agile, task-diverse movements.

**Iterated sampling-based online reinforcement learning.** Inspired by early successes of iterated sampling for offline and imitation decision making (Janner et al., 2022; Chi et al., 2023), many works have adapted diffusion-based policy parameterization to online reinforcement learning to encourage diverse action distribution (Yang et al., 2023; Li et al., 2024; Ishfaq et al., 2025; Celik et al., 2025). DACER introduces a diffusion actor–critic with an entropy regulator to stabilize policy learning and maintain exploration (Wang et al., 2024). Given unknown target distributions, several studies utilize the learned value function to regularize the policy learning (Ding et al., 2024; Dong et al., 2025; Jain et al., 2025). QSM matches the score of diffusion policy with the gradient of the Q-function (Psenka et al., 2024). SDAC introduces a Q-reweighted score matching function to avoid unstable training of backpropagating gradients through the diffusion chain (Ma et al., 2025). Recent works also employ flow-based policy in KL-constrained policy optimization (Lv et al., 2025; McAllister et al., 2025). These methods typically use standard Gaussian as a general initial distribution for the primary goal of multi-modal policy learning. The uninformative, isotropic bases can hinder scalability of policy learning in high-dimensional continuous control.

## 3 PRELIMINARIES

### 3.1 HIGH-DIMENSIONAL CONTINUOUS CONTROL

In this paper, we formalize the control of high-dimensional dynamical system as a infinite horizon Markov decision process (MDP) defined by the tuple $\mathcal{M} := \{\mathcal{S}, \mathcal{A}, \gamma, f, r, \rho\}$, where $\mathcal{S} \subset \mathbb{R}^{|\mathcal{S}|}$ is

the state space, $\mathcal{A} \subset [-1, 1]^{|\mathcal{A}|}$ is the action space, $\gamma$ is the discount factor, $f := \mathcal{S} \times \mathcal{A} \to \mathcal{P}(\mathcal{S})$ is the transition probability to $s' \in \mathcal{S}$ when being in $s \in \mathcal{S}$ and executing $a \in \mathcal{A}$, $r := \mathcal{S} \times \mathcal{A} \to \mathbb{R}$ is the reward function, and $\rho := \mathcal{S} \to \mathcal{P}(\mathcal{S})$ is the initial state distribution. The MDP starts from an initial state $s_0$ sampled from $\rho$, and proceeds with actions sampled from a policy $\pi := \mathcal{S} \to \mathcal{P}(\mathcal{A})$.

Our goal is to optimize the policy parameters to maximize the discounted cumulative reward:

$$J(\pi) = \mathbb{E}_\pi \sum_{h=0}^{\infty} \gamma^h r(s_h, a_h). \tag{1}$$

Compared with low-dimensional dynamical systems, controlling high-dimensional dynamics is substantially more challenging. We use the human musculoskeletal system as a motivating example to illustrate the difficulties of high-dimensional continuous control.

**High dimensionality.** Full-body human locomotion integrates rich sensory feedback with more than 600 muscles. Unlike robotic arms or quadrupeds, which typically operate within state and action spaces on the order of tens, the human musculoskeletal system features state and action spaces that are orders of magnitude larger. The size of the state-action space grows rapidly with dimension, leading to pronounced "curse-of-dimensionality" effects (Köppen, 2000). This demands expressive models and substantial informative data to reliably map high-dimensional states and actions to control performance.

**Over-actuation.** The number of biological actuators far exceeds the system's degrees of freedom (DoFs): many joints can be actuated by multiple muscles, and identical joint torques can arise from numerous activation patterns. This redundancy enlarges the feasible action set and complicates exploration and credit assignment, as multiple action sequences can yield indistinguishable kinematics but different internal forces and costs (Valero-Cuevas et al., 2015).

## 3.2 ACTOR-CRITIC ONLINE REINFORCEMENT LEARNING

Online reinforcement learning typically employs actor-critic framework, where the Q-function $Q^\pi(s, a)$ represents the value of state-action pair $(s, a)$ under policy $\pi$:

$$Q^\pi(s, a) = \mathbb{E}_\pi \left[ \sum_{h=0}^{\infty} \gamma^h (r(s_h, a_h) \middle| s_0 = s, a_0 = a \right], \tag{2}$$

The value function and the policy can be iteratively learned via a two-step scheme: policy evaluation and policy improvement. During policy evaluation, the Q-function is updated by Bellman equation operator $\mathcal{T}^\pi$ from any function $Q$, which converges to $Q^\pi$ when the operation number goes to infinity:

$$\mathcal{T}^\pi Q(s_h, a_h) \triangleq r(s_h, a_h) + \gamma \mathbb{E}_{a_{h+1} \sim \pi} \left[ Q(s_{h+1}, a_{h+1}) \right], \tag{3}$$

where the transition data tuples $(s_h, a_h, r(s_h, a_h), s_{h+1})$ are collected by interacting with the environment and stored in the replay buffer $\mathcal{B}$. To enhance the stability of Q-function update, two or more Q-functions are learned with separate parameters, where we use the minimum estimation to compute the regression target. In high-dimensional continuous control setting, the transition tuples are often collected from large number of parallel environments for better time efficiency.

In policy improvement, the policy parameters $\theta$ can be updated via optimizing the Q-function:

$$\pi_{\text{new}} = \arg\max_\pi \mathbb{E}_{s, a \sim \pi} Q^{\pi_{\text{old}}}(s, a). \tag{4}$$

In practice, the Q-function and the policy are typically parameterized by neural network $Q_\phi$ and $\pi_\theta$, and optimized by minimizing the following loss functions with gradient descent:

$$\mathcal{L}_Q(\phi) = \mathbb{E}_{(s, a, s') \sim \mathcal{B}} \left[ (r(s, a) + \gamma \mathbb{E}_{s' \sim f, a' \sim \pi} [Q_\phi(s', a')] - Q_\phi(s, a))^2 \right], \tag{5}$$

$$\mathcal{L}_\pi(\theta) = \mathbb{E}_{s, a \sim \pi_\theta} [-Q_\phi(s, a)]. \tag{6}$$

## 3.3 FLOW MATCHING

Flow matching is a simulation-free method for generative modeling that learns a probability flow directly by matching velocity fields along continuous-time probability paths. Let $p^{(0)}(x_0)$ and

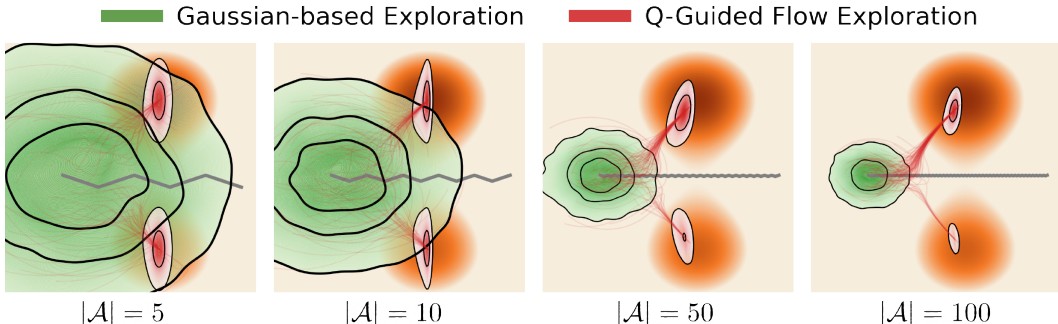

| Gaussian-based Exploration | Q-Guided Flow Exploration |
|---|---|

$|\mathcal{A}| = 5$   $|\mathcal{A}| = 10$   $|\mathcal{A}| = 50$   $|\mathcal{A}| = 100$

Figure 1: **Exploration behavior across increasing action dimensionality.** The gray polyline depicts a planar kinematic chain with $|\mathcal{A}|$ degrees of freedom. The orange background (darker is higher) visualizes the state–action value $Q$. Green contours show the end-effector distribution induced by an undirected Gaussian proposal over joint angles, whose exploratory reach collapses as $|\mathcal{A}|$ increases. Red streamlines/contours depict Q-guided probability flows that transport probability mass from the Gaussian proposal toward high-value modes, sustaining directed exploration in high dimensions.

$p^{(1)}(\boldsymbol{x}_1)$ denote the source and target distributions over $\mathbb{R}^d$ respectively. Flow matching considers a continuous-time probability path $\{p^{(t)}(\boldsymbol{x})\}_{t\in[0,1]}$ that evolves smoothly from $p^{(0)}(\boldsymbol{x}_0)$ to $p^{(1)}(\boldsymbol{x}_1)$. This evolution is governed by a velocity field $\boldsymbol{v}^{(t)}(\boldsymbol{x})$, such that the density $p^{(t)}$ satisfies the continuity equation:

$$\frac{dp^{(t)}(\boldsymbol{x})}{dt} + \nabla \cdot [p^{(t)}(\boldsymbol{x})\boldsymbol{v}^{(t)}(\boldsymbol{x}))] = 0. \tag{7}$$

Assuming the path $p^{(t)}$ is regular enough, one can define a flow map $\boldsymbol{\phi}^{(t)}$ through the following Ordinary Differential Equation (ODE):

$$\frac{d\boldsymbol{\phi}^{(t)}(\boldsymbol{x})}{dt} = \boldsymbol{v}^{(t)}(\boldsymbol{\phi}^{(t)}(\boldsymbol{x})), \quad \boldsymbol{\phi}^{(0)}(\boldsymbol{x}) = \boldsymbol{x}. \tag{8}$$

We denote $\boldsymbol{\phi}^{(t)}(\boldsymbol{x})$ as $\boldsymbol{x}^{(t)}$. Flow matching seeks to learn an approximate velocity field $\boldsymbol{v}_w(\boldsymbol{x},t)$, parameterized by a neural network, that induces a flow transporting $p_0$ to $p_1$. To train $\boldsymbol{v}_w$, flow matching minimizes the expected squared error between the model and a reference (or target) velocity field, which is tractable when conditioned on samples from the target distribution (Lipman et al., 2023):

$$\mathcal{L}_{\text{CFM}} = \mathbb{E}_{\substack{t\sim\mathcal{U}([0,1]) \\ \boldsymbol{x}^{(1)}\sim p^{(1)}(\cdot) \\ \boldsymbol{x}^{(t)}\sim p^{(t)}(\cdot|\boldsymbol{x}^{(1)})}} \left\| \boldsymbol{v}_w(\boldsymbol{x}^{(t)},t) - \boldsymbol{v}^{(t)}(\boldsymbol{x}^{(t)}|\boldsymbol{x}^{(1)}) \right\|^2 \tag{9}$$

## 4 VANISHING EFFECTIVENESS OF UNDIRECTED EXPLORATION

To enable exploration during interaction with the environment, the policy $\pi$ is often designed to be a stochastic distribution (Haarnoja et al., 2018) or perturbed with undirected, isotropic noises (Schulman et al., 2017). Gaussian distribution is a commonly used choice for policy and noise parameterization for its simplicity and tractable likelihood computation. The policy that uses Gaussian for exploration follows the generalized form as:

$$\pi(\boldsymbol{a}|\boldsymbol{s}) = \mathcal{N}(\mu(\boldsymbol{s}), \sigma^2(\boldsymbol{s})), \tag{10}$$

where $\mu := \mathbb{R}^{|\mathcal{S}|} \to \mathbb{R}^{|\mathcal{A}|}$ and $\sigma := \mathbb{R}^{|\mathcal{S}|} \to \mathbb{R}^{|\mathcal{A}|}$ are mean and (typically) diagonal standard deviation functions of each Gaussian action distribution. In this section, we demonstrate that the undirected stochasticity leads to vanishing exploration in high-dimensional continuous control by the following case analysis:

**Case analysis: vanishing exploration in high DoF settings.** Consider a planar kinematic chain with $|\mathcal{A}|$ degrees-of-freedom (i.e. $|\mathcal{A}|$ revolute joints and a terminal link) in 2D, where each link has length $l_i = L/|\mathcal{A}|$. Under i.i.d. zero-mean joint-angle perturbations with fixed variance, the end-effector position variance scales as $O(\frac{1}{|\mathcal{A}|})$; equivalently, it decays proportionally to $\frac{1}{|\mathcal{A}|}$ as $|\mathcal{A}|$ grows. (See **Appendix A.1** for the proof.)

We visualize the vanishing exploration in Figure 1. When the action dimension is moderate ($\mathcal{A} \leq 10$), Gaussian-based exploration suffices to find informative samplings with high values. However, the diversity of the Gaussian-based exploration collapses as the system complexity grows, leading to uninformative sampling behavior. Related observations of vanishing exploration for over-actuated systems have also been reported in previous works (Schumacher et al., 2023b). These findings motivate directed exploration mechanisms rather than relying on isotropic perturbations.

---

**Algorithm 1** Q-guided Flow Exploration (QFLEX)

---

**Input:** Initialized parameters $\theta, w, \phi$, gradient step number $N$, initial gradient step size $\eta$, learning rates $\lambda_\phi, \lambda_\theta, \lambda_w$

1: **for** $h = 1, 2, \cdots$ **do**
2: $\quad \boldsymbol{a}_h \sim \pi_{\theta,w}^{(1)}(\cdot|\boldsymbol{s}_h)$
3: $\quad \boldsymbol{s}_{h+1} \sim p(\cdot|\boldsymbol{a}_t^0, \boldsymbol{s}_h)$
4: $\quad \mathcal{B} \leftarrow \mathcal{B} \bigcup \{\boldsymbol{s}_h, \boldsymbol{a}_h, r_h, \boldsymbol{s}_{h+1}\}$
5: $\quad \phi \leftarrow \phi - \lambda_\phi \nabla_\phi \mathcal{L}_Q(\phi)$ $\qquad\qquad\qquad\qquad\qquad$ ▷ Eq. (5)
6: $\quad \theta \leftarrow \theta - \lambda_\theta \nabla_\theta \mathcal{L}_\pi(\theta)$ $\qquad\qquad\qquad\qquad\qquad$ ▷ Eq. (6)
7: $\quad \boldsymbol{a}^{(0)} \sim \pi_\theta^{(0)}(\cdot|\boldsymbol{s}_h)$
8: $\quad$ **for** $n = 1$ to $N$ **do**
9: $\quad\quad \boldsymbol{a}^{(\frac{n}{N})} \leftarrow \boldsymbol{a}^{(\frac{n-1}{N})} + \bar{\eta} \nabla_{\boldsymbol{a}} Q_\phi(\boldsymbol{s}_h, \boldsymbol{a}^{(\frac{n-1}{N})})$ $\qquad$ ▷ Q-guided flow construction
10: $\quad$ **end for**
11: $\quad w \leftarrow w - \lambda_w \nabla_w \mathcal{L}_{\boldsymbol{v}}(w)$ $\qquad\qquad\qquad\qquad\qquad$ ▷ Eq. (17)
12: **end for**

---

## 5 FLOW-BASED POLICY FOR SCALABLE EXPLORATION

In this section, we first demonstrate how policy improvement can be achieved by sampling from a probability flow guided by the learned state-action value function. Then we introduce QFLEX , an efficient online RL method for scalable exploration in high-dimensional continuous control.

### 5.1 POLICY IMPROVEMENT VIA VALUE-GUIDED FLOW

Since undirected exploration vanishes in high-dimensional action space, a suitable strategy to explore is to take the "best" action under current experience. Given policy $\pi_{\text{old}}$ and the Q-function $Q^{\pi_{\text{old}}}$, the policy improvement procedure in Eq. (4) seeks a new policy $\pi_{\text{new}}$ that maximizes the expectation of $Q^{\pi_{\text{old}}}$. We denote the policy learned by minimizing Eq. (6) as $\pi^{(0)}$. In practice, $\pi^{(0)}$ often deviates from $\pi_{\text{new}}$ due to factors such as restrictive parameterization or insufficient optimization. To bridge this gap, we construct a Q-guided velocity field that transports $\pi^{(0)}$ towards $\pi_{\text{new}}$:

$$\frac{d\boldsymbol{a}^{(t)}}{dt} = \boldsymbol{v}_Q^{(t)}(\boldsymbol{a}^{(t)}; \boldsymbol{s}) = \boldsymbol{M} \nabla_{\boldsymbol{a}} Q^{\pi_{\text{old}}}(\boldsymbol{s}, \boldsymbol{a}), \quad \boldsymbol{a}^{(0)} \sim \pi^{(0)}(\cdot|\boldsymbol{s}), \quad (11)$$

where $\boldsymbol{M}$ is any positive definite preconditioner that that rescales and reorients the raw action-gradient. Defining the advantage of $\pi^{(t)}$ over $\pi^{(0)}$ as:

$$F(t; \boldsymbol{s}) = \mathbb{E}_{\boldsymbol{a} \sim \pi^{(t)}(\cdot|\boldsymbol{s})} \left[ Q^{\pi_{\text{old}}}(\boldsymbol{s}, \boldsymbol{a}) - \mathbb{E}_{\boldsymbol{a}' \sim \pi^{(0)}(\cdot|\boldsymbol{s})} [Q^{\pi_{\text{old}}}(\boldsymbol{s}, \boldsymbol{a}')] \right]. \quad (12)$$

Under mild regularity assumptions, the transformed policy $\pi^{(t)}(\cdot|\boldsymbol{s}) = \phi_{\boldsymbol{s}}^{(t)}(\pi_\theta(\cdot|\boldsymbol{s}))$ constitutes a valid policy-improvement flow which increases the expected state–action value.

**Proposition 1.** *Assuming $Q^{\pi_{\text{old}}}$ is once continuously differentiable with locally Lipschitz $\nabla_{\boldsymbol{a}} Q^{\pi_{\text{old}}}$, $\boldsymbol{M}$ has bounded operator norm $\|\boldsymbol{M}\|$ and $\|\nabla_{\boldsymbol{a}} Q^{\pi_{\text{old}}}\|_{\boldsymbol{M}}$ is intergrable under $\pi^{(t)}(\cdot|\boldsymbol{s})$ for relevant $t$. Then the map $t \to F(t; \boldsymbol{s})$ is monotone nondecreasing, i.e., $\frac{d}{dt} F(t; \boldsymbol{s}) \geq 0$. (See **Appendix A.2** for the proof.)*

Figure 1 demonstrates that the Q-guided flow consistently steers actions toward high-value regions across action dimensionalities, enabling directed exploration and yielding more informative samples.

## 5.2 Q-GUIDED FLOW EXPLORATION

As summarized in Algorithm 1, we embed the above results into an actor-critic online RL routine and introduce Q-guided Flow Exploration (QFLEX), which explores high-dimensional action spaces via sampling from the Q-guided conditional normalizing flow in Eq. (11). We parameterize the flow-based policy via a Gaussian initializer $\pi_\theta^{(0)}(\boldsymbol{a}|\boldsymbol{s})$ and state-dependent velocity field $\boldsymbol{v}_w(\boldsymbol{a}|t, \boldsymbol{s}, \boldsymbol{a}^{(t)})$. Starting from initial samples drawn from Gaussian policy, QFLEX transform actions following the learned vector field by solving the ODE:

$$\pi_{\theta,w}^{(1)}(\boldsymbol{a}|\boldsymbol{s}, \boldsymbol{a}^{(0)}) = \boldsymbol{a}^{(0)} + \int_0^1 \boldsymbol{v}_w(t, \boldsymbol{s}, \boldsymbol{a}^{(t)})dt, \quad \boldsymbol{a}^{(0)} \sim \pi_\theta^{(0)}(\cdot|\boldsymbol{s}). \tag{13}$$

The training of QFLEX proceeds as follows:

**Update of Q-function and Gaussian policy.** At each training iteration, QFLEX collects trajectories into replay buffer $\mathcal{B}$ by sampling from the flow-induced policy $\pi_{\theta,w}^{(1)}$ (line 2-4). The Q-function and the Gaussian policy are updated according to the standard policy iteration and policy improvement steps (line 5-6). Since the sample efficiency of QFLEX hinges on the quality of the learned Q-function, we employ batch normalization within the Q-network to normalize state–action batches and stabilize optimization (Bhatt et al., 2024). This stabilization allows us to dispense with a target Q-network and to train with a low update-to-data ratio, yielding more efficient Q-learning.

**Q-guided flow construction.** Starting from samples of $\pi_\theta^{(0)}$, we adopt identity matrix $\boldsymbol{I}$ as the preconditioner of the Q-guided velocity field, which corresponds to Euclidean steepest ascent in action space. We then construct the Q-guided flow by taking $N$ finite gradient-ascent steps on the differentiable Q-function, where the transported actions $\boldsymbol{a}^{(1)}$ are treated as samples from the target distribution $\pi^{(1)}$ (line 7-9). Because the Q-network's gradients can be poorly behaved outside the admissible action domain, updates near the boundary may push actions outside $[-1, 1]^{|\mathcal{A}|}$. Thus a fixed step size $\eta$ can destabilize learning. To mitigate this, we cap each update using the $l_2$-diameter of the action space:

$$\boldsymbol{a}^{(\frac{n}{N})} \leftarrow \boldsymbol{a}^{(\frac{n-1}{N})} + \bar\eta \nabla_{\boldsymbol{a}} Q_\phi(\boldsymbol{s}_h, \boldsymbol{a}^{(\frac{n-1}{N})}), \quad \bar\eta = \min\left(\eta, \frac{2\sqrt{|\mathcal{A}|}}{\left\|\nabla_{\boldsymbol{a}} Q_\phi(\boldsymbol{s}_h, \boldsymbol{a}^{(\frac{n-1}{N})})\right\|}\right). \tag{14}$$

The truncated step size bounds the per-iteration displacement, enabling stable, valid exploration within the action space.

**Update of Q-guided velocity field.** Given target $\boldsymbol{a}^{(1)}$ and source sample $\boldsymbol{a}^{(0)}$ from Gaussian policy $\pi_\theta^{(0)}$, we specify the optimal transport conditional probability path and its target velocity field as:

$$p^{(t)}(\boldsymbol{a}^{(t)}|\boldsymbol{s}, \boldsymbol{a}^{(0)}, \boldsymbol{a}^{(1)}) = \delta\left(\boldsymbol{a}^{(t)} - \left[(1-t)\boldsymbol{a}^{(0)} + t\boldsymbol{a}^{(1)}\right]\right), \tag{15}$$

$$\boldsymbol{v}^{(t)}(\boldsymbol{a}^{(t)}|\boldsymbol{s}, \boldsymbol{a}^{(0)}, \boldsymbol{a}^{(1)}) = \boldsymbol{a}^{(1)} - \boldsymbol{a}^{(0)}, \tag{16}$$

where $\delta(\cdot)$ denotes the Dirac distribution. The velocity field $\boldsymbol{v}_w$ can be updated by optimizing the state-dependent conditional flow matching loss (line 11):

$$\mathcal{L}_{\boldsymbol{v}}(w) = \mathbb{E}_{\substack{t\sim\mathcal{U}([0,1]) \\ \boldsymbol{s},\boldsymbol{a}^{(0)}\sim\pi_\theta^{(0)}(\cdot|\boldsymbol{s}) \\ \boldsymbol{a}^{(1)}\sim\pi^{(1)}(\cdot|\boldsymbol{s},\boldsymbol{a}^{(0)}) \\ \boldsymbol{a}^{(t)}\sim p^{(t)}(\cdot|\boldsymbol{s},\boldsymbol{a}^{(0)},\boldsymbol{a}^{(1)})}} \left\|\boldsymbol{v}_w(t, \boldsymbol{s}, \boldsymbol{a}^{(t)}) - \boldsymbol{v}^{(t)}(\boldsymbol{a}^{(t)}|\boldsymbol{s}, \boldsymbol{a}^{(0)}, \boldsymbol{a}^{(1)})\right\|^2 \tag{17}$$

Compared with diffusion-based online RL methods that initialize from a fixed standard Gaussian, QFLEX maintains a *learnable* source distribution. This yields informative initialization points for transport toward the target distribution and substantially easing the learning of high-performing flow-based policies. In contrast to approaches that rely on dimensionality reduction, QFLEX preserves

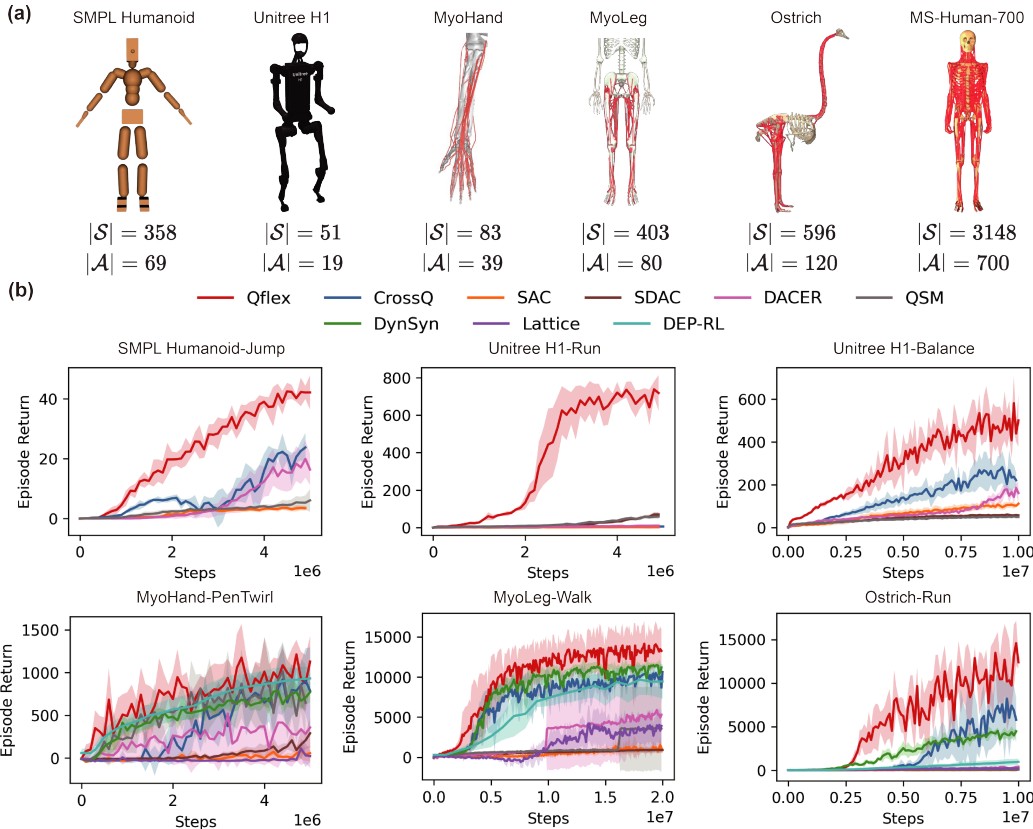

Figure 2: **Control over high-dimensional control benchmarks.** (a) Morphologies and state-action dimensions of evaluated benchmarks. (b) Learning curve of algorithms. Results show mean performances with one standard deviation of 5 independent runs. Baselines in the second row are run only on musculoskeletal benchmarks.

the full flexibility of high-dimensional dynamical systems by exploring the *original* action space, thereby facilitating agile, complex motor control.

Although Algorithm 1 presents a minimalist instantiation, QFLEX readily extends to various RL frameworks and exploration regimes. The flow-based policy parameterization permits direct simulation of policy likelihoods via the instantaneous change of variables (Chen et al., 2018), making QFLEX naturally compatible with KL-constrained policy optimization (Schulman et al., 2015; 2017) and maximum-entropy RL (Haarnoja et al., 2017). Moreover, geometry-aware or curvature-adaptive choices of the preconditioner $M$ (e.g. natural-gradient or Newton-type updates) can induce more structured exploration to accelerate search. We leave a systematic study of these design choices to future work.

## 6   EXPERIMENT

In this section, we present a comprehensive evaluation of QFLEX for high-dimensional continuous control. We first compare QFLEX against extensive online RL baselines on simulated benchmarks. Then we demonstrate its control performance on a 700-actuator human musculoskeletal model executing agile, full-body movements. Finally, we analyze QFLEX's behavior to assess its scalability in exploration. For all experiments, we construct the Q-guided flow by $N = 20$ gradient steps with initial step size $\eta = 0.01$. The ODE in Eq. (13) is solved with a naive Euler integrator by 20 discrete steps with timestep $\Delta t = 0.05$. Our code and video results can be found in our project page: https://lnsgroup.cc/research/Qflex.

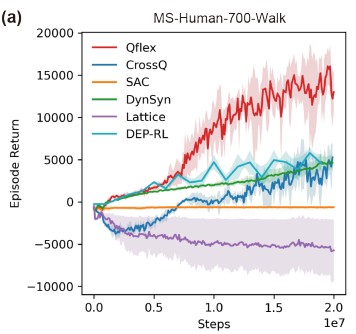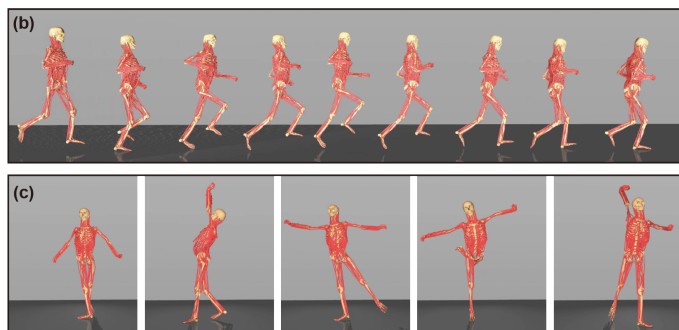

Figure 3: **Control over full-body human musculoskeletal system.** (a) Learning efficiency over walking control of MS-Human-700. Results show mean performances with one standard deviation of 5 independent runs. (b) Learned behavior of whole-body running. (c) Learned behavior of ballet dancing.

## 6.1 CONTROL OVER HIGH-DIMENSIONAL SIMULATED BENCHMARKS

We evaluate on a diverse suite of simulated high-dimensional continuous-control benchmarks: **(1) SMPL Humanoid–Jump** (Tirinzoni et al., 2025), which controls a humanoid agent based on the SMPL skeleton (Loper et al., 2023) to execute jumps; **(2) Unitree H1–Run/Balance** (Sferrazza et al., 2024), which controls a Unitree H1 humanoid[1] to run forward or maintain balance on an unstable platform; **(3) MyoHand–PenTwirl / MyoLeg–Walk** (Caggiano et al., 2022), which controls a hand musculoskeletal system to twirl a pen and a lower-body musculoskeletal system to walk; and **(4) Ostrich–Run** (La Barbera et al., 2021), which controls an ostrich musculoskeletal system to run. The state and action spaces for all tasks are summarized in Figure 2 (a).

We compare QFLEX to representative online RL baselines: **(1) Gaussian-based:** CrossQ (Bhatt et al., 2024), SAC (Haarnoja et al., 2018); **(2) Diffusion-based:** SDAC (Ma et al., 2025), DACER (Wang et al., 2024), QSM (Psenka et al., 2024); and **(3) High-dimensional musculoskeletal control:** DynSyn (He et al., 2024), Lattice (Chiappa et al., 2023a), DEP-RL (Schumacher et al., 2023b).

As shown in Figure 2 (b), QFLEX demonstrates consistently superior learning efficiency across all benchmarks. The performance gap widens with increasing action dimensionality and over-actuation, indicating scalable exploration behavior.

## 6.2 CONTROL OVER FULL-BODY HUMAN MUSCULOSKELETAL SYSTEM

We employ QFLEX for locomotion control of MS-Human-700 (Zuo et al., 2024), a full-body musculoskeletal system with 206 joints and 700 muscle-tendon units. Its state–action dimensionality is more than five times that of the most complex benchmark in the previous subsection (Ostrich–Run). As shown in Figure 3 (a), QFLEX exhibits high-learning efficiency and strong scalability over whole-body walking control, outperforms existing high-dimensional musculoskeletal control baselines by a large margin *without* dimension reduction.

We further deploy QFLEX on two challenging skills—running and ballet dancing—that, to our knowledge, have not previously been demonstrated on a 700-actuator full-body system. In Figure 3 (b), QFLEX enables rapid high-dimensional sensorimotor coordination,

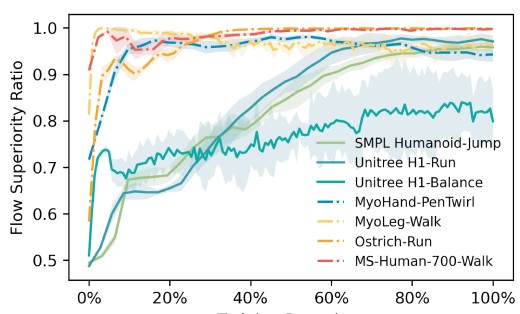

Figure 4: **Sample quality between QFLEX and source Gaussian policy during training.** Over-actuated musculoskeletal control tasks are denoted as dash-dotted lines.

---

[1]https://github.com/unitreerobotics/unitree_ros

achieving a stable running gait. In Figure 3 (c), QFLEX successfully imitates a ballet routine featuring complex whole-body sequences with single-foot spins and balance. By exploring in the *original* action space, QFLEX fully leverages the flexibility of high-dimensional dynamical systems, enabling agile and complex motion control.

### 6.3 ALGORITHM ANALYSIS

We examine QFLEX's efficiency by analyzing its learning dynamics. To compare sampling quality between the flow-based policy and the Gaussian reference, we track the *flow superiority ratio* during training, which is the proportion of states in a minibatch for which

$$Q(\boldsymbol{s}, \pi_{\theta,w}^{(1)}(\cdot|\boldsymbol{s})) > Q(\boldsymbol{s}, \pi_{\theta}^{(0)}(\cdot|\boldsymbol{s})). \tag{18}$$

As shown in Figure 4, the flow-based policy consistently yields higher state-action values than Gaussian exploration, and this advantage strengthens over the course of training. Notably, the superiority ratio is substantially higher on musculoskeletal control tasks than on torque-controlled benchmarks, underscoring the importance of value-aligned exploration in high-dimensional, over-actuated settings.

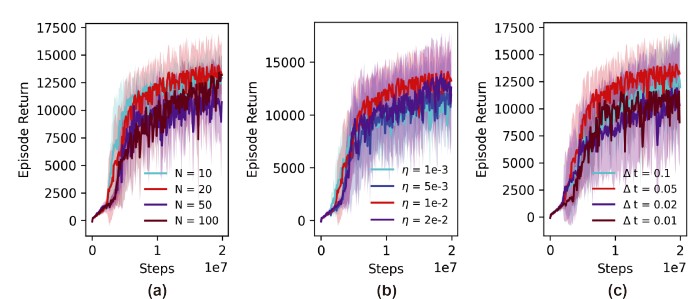

Figure 5: **Ablation study over hyperparameters of** QFLEX. (a) Gradient steps $N$. (b) Step size $\eta$. (c) Euler solving timestep $\Delta t$.

On MyoLeg-Walk task, we further perform a sensitivity study over QFLEX's hyperparameters: number of gradient steps $N$, step size $\eta$ and Euler solving timestep $\Delta t$. Figure 5 shows broadly comparable learning performance across a reasonable range of these choices.

## 7 CONCLUSION

In this paper, we introduce QFLEX, a scalable online RL method for efficient exploration in high-dimensional continuous control. Our method conducts directed exploration by sampling from a Q-guided probability flow with policy-improvement guarantees, yielding superior learning efficiency over representative online RL baselines across benchmarks characterized by high dimensionality and over-actuation. QFLEX further demonstrates agile, complex motion control on a full-body musculoskeletal model with 700 actuators, achieving high efficiency and strong scalability in truly high-dimensional settings. Our analysis shows that value-aligned exploration in QFLEX surpasses undirected sampling strategies in high-dimensional regimes, which is readily extensible to a variety of online RL frameworks and exploration settings.

#### ACKNOWLEDGMENTS

This work is supported by STI 2030-Major Projects 2022ZD0209400, Beijing Academy of Artificial Intelligence and Beijing Municipal Science & Technology Commission Z251100008125022, and AMD Research. Correspondence to: Yanan Sui (ysui@tsinghua.edu.cn).

**Ethics statement.** This work follows the ICLR Code of Ethics. We considered the potential ethical and societal impacts of this work. No human or animal subjects were directly involved. We report limitations and assumptions transparently and strive to promote beneficial and responsible use of this work.

**Reproducibility statement.** We provide an anonymous codebase, full hyperparameters, and exact evaluation protocols to enable faithful replication.

**LLM usage statement.** We used a large language model solely for language polishing.

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

# A   THEORETICAL PROOFS

## A.1   PROOF OF CASE ANALYSIS

**Case analysis: vanishing exploration in high DoF settings.** Consider a planar kinematic chain with $|\mathcal{A}|$ degrees-of-freedom (i.e. $|\mathcal{A}|$ revolute joints and a terminal link) in 2D, where each link has length $l_i = L/|\mathcal{A}|$. Under i.i.d. zero-mean joint-angle perturbations with fixed variance, the end-effector position variance scales as $O(\frac{1}{|\mathcal{A}|})$; equivalently, it decays proportionally to $\frac{1}{|\mathcal{A}|}$ as $|\mathcal{A}|$ grows.

*Proof.* We denote the position of the end-effector as $\boldsymbol{x} = (x, y)$. The forward kinematics of the system can be expressed as a function:

$$\boldsymbol{x} = f(\boldsymbol{\varphi}), \tag{19}$$

where $\boldsymbol{\varphi} = (\varphi_1, \cdots, \varphi_{|\mathcal{A}|})$ is the system joint positions. For small noise, we can use a first-order Taylor expansion of $f(\boldsymbol{\varphi})$ around the current joint position $\bar{\boldsymbol{\varphi}}$:

$$\boldsymbol{x} \approx f(\bar{\boldsymbol{\varphi}}) + J(\bar{\boldsymbol{\varphi}})\delta\boldsymbol{\varphi}, \tag{20}$$

where $J(\bar{\boldsymbol{\varphi}})$ is the Jacobian matrix of the forward kinematics with respect to $\bar{\boldsymbol{\varphi}}$, and $\delta\boldsymbol{\varphi} = (\delta\varphi_1, \cdots, \delta\varphi_{|\mathcal{A}|})$ with $\mathrm{Var}(\delta\varphi_i) = \sigma_i^2$. The covariance matrix of $\delta\boldsymbol{\varphi}$ is $\Sigma_{\boldsymbol{\varphi}} = \sigma^2 \boldsymbol{I}$. Therefore the covariance of the end position $\boldsymbol{x}$ is given by:

$$\Sigma_{\boldsymbol{x}} = J(\bar{\boldsymbol{\varphi}})\Sigma_{\boldsymbol{\varphi}}J(\bar{\boldsymbol{\varphi}})^T = \sum_{i=1}^{|\mathcal{A}|} \sigma_i^2 \|J_{:,i}\| \leq \sigma_{\max}^2 \sum_{i=1}^{|\mathcal{A}|} \|J_{:,i}\|, \tag{21}$$

where $\|J_{:,i}\|$ is the norm of the $i$-th column of the Jacobian matrix $J$, and $\sigma_{\max} = \max_i \sigma_i$ . Where we can extract the end position variance as the trace of $\Sigma_{\boldsymbol{x}}$:

$$\mathrm{Var}(\boldsymbol{x}) = \mathrm{Tr}(\Sigma_{\boldsymbol{x}}) \tag{22}$$

For a planar $|\mathcal{A}|$-link system where each link $l_i = L/|\mathcal{A}|$, the Jacobian entries are influenced by these link lengths, and the trace term can be approximated as:

$$\mathrm{Tr}(\sum_{i=1}^{|\mathcal{A}|} \|J_{:,i}\|) \approx |\mathcal{A}| \left(\frac{L^2}{|\mathcal{A}|^2}\right) = \frac{L^2}{|\mathcal{A}|}, \tag{23}$$

which leads to the total variance of the end-effector as:

$$\mathrm{Var}(\boldsymbol{x}) = \frac{\sigma_{\max}^2 L^2}{|\mathcal{A}|} \tag{24}$$

$\square$

## A.2   PROOF OF PROPOSITION 1

**Proposition 1.** *Assuming $Q^{\pi_{old}}$ is once continuously differentiable with locally Lipschitz $\nabla_{\boldsymbol{a}} Q^{\pi_{old}}$, $M$ has bounded operator norm $\|M\|$ and $\|\nabla_{\boldsymbol{a}} Q^{\pi_{old}}\|_M$ is intergrable under $\pi^{(t)}(\cdot|\boldsymbol{s})$ for relevant $t$. Then the map $t \to F(t; \boldsymbol{s})$ is monotone nondecreasing, i.e., $\frac{d}{dt} F(t; \boldsymbol{s}) \geq 0$.*

*Proof.* Let $\boldsymbol{a}^{(t)} = \phi_{\boldsymbol{s}}^{(t)}(\boldsymbol{a}^{(0)})$. Then we can reparameterize $F$ as

$$F(t; \boldsymbol{s}) = \mathbb{E}_{\boldsymbol{a} \sim \pi^{(0)}(\cdot|\boldsymbol{s})} \left[ Q^{\pi_{old}}(\boldsymbol{s}, \phi_{\boldsymbol{s}}^{(t)}(\boldsymbol{a})) - \mathbb{E}_{\boldsymbol{a}' \sim \pi^{(0)}(\cdot|\boldsymbol{s})}[Q^{\pi_{old}}(\boldsymbol{s}, \boldsymbol{a}')] \right] \tag{25}$$

The differentiate under the expectation is

$$\frac{d}{dt}F(t; \boldsymbol{s}) = \frac{d}{dt}\mathbb{E}_{\boldsymbol{a}\sim\pi^{(0)}(\cdot|\boldsymbol{s})}\left[Q^{\pi_{\text{old}}}(\boldsymbol{s}, \boldsymbol{\phi}_{\boldsymbol{s}}^{(t)}(\boldsymbol{a}) - \mathbb{E}_{\boldsymbol{a}'\sim\pi^{(0)}(\cdot|\boldsymbol{s})}[Q^{\pi_{\text{old}}}(\boldsymbol{s}, \boldsymbol{a}')]\right] \tag{26}$$

$$= \frac{d}{dt}\mathbb{E}_{\boldsymbol{a}\sim\pi^{(0)}(\cdot|\boldsymbol{s})}\left[Q^{\pi_{\text{old}}}(\boldsymbol{s}, \boldsymbol{\phi}_{\boldsymbol{s}}^{(t)}(\boldsymbol{a}))\right] \tag{27}$$

$$= \mathbb{E}_{\boldsymbol{a}\sim\pi^{(0)}(\cdot|\boldsymbol{s})}\left[\nabla_{\boldsymbol{a}}Q^{\pi_{\text{old}}}(\boldsymbol{s}, \boldsymbol{\phi}_{\boldsymbol{s}}^{(t)}(\boldsymbol{a}))^\top \frac{d}{dt}\boldsymbol{\phi}_{\boldsymbol{s}}^{(t)}(\boldsymbol{a})\right] \tag{28}$$

$$= \mathbb{E}_{\boldsymbol{a}\sim\pi^{(0)}(\cdot|\boldsymbol{s})}\left[\nabla_{\boldsymbol{a}}Q^{\pi_{\text{old}}}(\boldsymbol{s}, \boldsymbol{\phi}_{\boldsymbol{s}}^{(t)}(\boldsymbol{a}))^\top M \nabla_{\boldsymbol{a}}Q^{\pi_{\text{old}}}(\boldsymbol{s}, \boldsymbol{\phi}_{\boldsymbol{s}}^{(t)}(\boldsymbol{a}))\right] \tag{29}$$

$$= \mathbb{E}_{\boldsymbol{a}\sim\pi^{(t)}(\cdot|\boldsymbol{s})}\left[\nabla_{\boldsymbol{a}}Q^{\pi_{\text{old}}}(\boldsymbol{s}, \boldsymbol{a})^\top M \nabla_{\boldsymbol{a}}Q^{\pi_{\text{old}}}(\boldsymbol{s}, \boldsymbol{a})\right] \tag{30}$$

$$= \mathbb{E}_{\boldsymbol{a}\sim\pi^{(t)}(\cdot|\boldsymbol{s})}\left[\|\nabla_{\boldsymbol{a}}Q^{\pi_{\text{old}}}(\boldsymbol{s}, \boldsymbol{a})\|_M^2\right] \geq 0, \tag{31}$$

where Eq. (28) follows the derivative chain rule, and Eq. (30) is derived by reparameterization. $\square$

## B    EXPERIMENTAL DETAILS

### B.1    ALGORITHM IMPLEMENTATION

We implement QFLEX on the JAX platform (Bradbury et al., 2018). Specifically, the neural networks are implemented using Haiku[2] with parameters optimized with Optax[3].

For the implementation of SAC, DACER and QSM, we refer to DACER-Diffusion-with-Online-RL[4] in the official code repository of DACER, which provide efficient JAX-based implementation of SAC and diffusion-based online RL baselines.

For the implementation of SDAC, we directly use the official repository[5], which provides JAX-based implement based on DACER repository.

For the implementation of CrossQ, we refer to the official repository[6], and reproduce a JAX-based implementation to improve the time efficiency of training.

For the implementation of DynSyn, we directly use the official repository[7], and use SAC as the RL backbone.

For the implementation of Lattice, we directly use the official repository[8], and use SAC as the RL backbone.

For the implementation of DEP-RL, we directly use the official repository[9], and use SAC as the RL backbone.

For all algorithms, we align the network parameters and learning rate with 1 gradient steps after each parallel sampling step. For the training of Lattice, we follow the default training setting of 8 gradient steps after 8 parallel sampling steps, which we consider comparable to other baselines on average. Otherwise we use the default hyperparameter in the original implementation. The full experimental details is listed in Table 1 and 2.

---

[2]https://github.com/google-deepmind/dm-haiku
[3]https://github.com/google-deepmind/optax
[4]https://github.com/happy-yan/DACER-Diffusion-with-Online-RL
[5]https://github.com/mahaitongdae/diffusion_policy_online_rl
[6]https://github.com/adityab/CrossQ
[7]https://github.com/Beanpow/DynSyn
[8]https://github.com/amathislab/lattice
[9]https://github.com/martius-lab/depRL

Table 1: Training details of each environments.

| System | Unitree H1 | SMPL Humanoid | MyoHand | MyoLeg | Ostrich | MS-Human-700 |
|---|---|---|---|---|---|---|
| Parallel number | 70 | 80 | 80 | 80 | 80 | 224 |
| Critic hidden layer | 3 | 3 | 3 | 3 | 3 | 3 |
| Critic hidden size | 256 | 256 | 256 | 256 | 256 | 1024 |
| Policy hidden layer | 3 | 3 | 3 | 3 | 3 | 3 |
| Policy hidden size | 256 | 256 | 256 | 256 | 256 | 1024 |
| Diffusion/flow hidden layer | 3 | 3 | 3 | 3 | 3 | 3 |
| Diffusion/flow hidden size | 256 | 256 | 256 | 256 | 256 | 1024 |

Table 2: Hyperparameter settings.

| | QFLEX | SDAC | DACER | QSM | CrossQ | SAC | DynSyn | DEP-RL | Lattice |
|---|---|---|---|---|---|---|---|---|---|
| Gradient steps | 1 | 1 | 1 | 1 | 1 | 1 | 1 | 1 | 8/8 |
| Discount | 0.99 | 0.99 | 0.99 | 0.99 | 0.99 | 0.99 | 0.99 | 0.99 | 0.99 |
| Batch size | 256 | 256 | 256 | 256 | 256 | 256 | 256 | 256 | 256 |
| Buffer size | 1e6 | 1e6 | 1e6 | 1e6 | 1e6 | 1e6 | 1e6 | 1e6 | 1e6 |
| Learning rate | 3e-4 | 3e-4 | 3e-4 | 3e-4 | 3e-4 | 3e-4 | 3e-4 | 3e-4 | 3e-4 |
| Optimizer | Adam | Adam | Adam | Adam | Adam | Adam | Adam | Adam | Adam |
| Diffusion/flow steps | 20 | 20 | 20 | 20 | - | - | - | - | - |
| Batch normalization decay | 0.99 | - | - | - | 0.99 | - | - | - | - |
| Batch normalization $\epsilon$ | 1e-5 | - | - | - | 1e-5 | - | - | - | - |
| Target policy entropy | - | $-0.9 \cdot |\mathcal{A}|$ | $-0.9 \cdot |\mathcal{A}|$ | - | $-|\mathcal{A}|$ | $-|\mathcal{A}|$ | $-|\mathcal{A}|$ | $-|\mathcal{A}|$ | $-|\mathcal{A}|$ |

## B.2 BENCHMARK IMPLEMENTATION

**SMPL-Humanoid-Jump.** We implement the benchmark using the `jump-2` task in the official Humenv repository[10] with provided reward function. The environment is wrapped to be compatible with Gymnasium environment make function.

**Unitree H1-Run/Balance.** We use `h1-run-v0` and `h1-balance_simple-v0` tasks in the official HumanoidBench repository[11] to implement the benchmark with provided reward function.

**MyoHand-PenTwirl/MyoLeg-Walk.** We use `myoHandPenTwirlRandom-v0` and `myoLegWalk-v0` tasks in the official MyoSuite repository[12] to implement the benchmarks with provided reward function.

**Ostrich-Run.** We implement the benchmarks using the `ostrich-run` task in the official OstrichRL repository[13] with provided reward function. The environment is wrapped to be compatible with Gymnasium environment make function.

**MS-Human-700-Walk.** We develop task environments with MS-Human-700 under Gymnasium. The full states (observations) and dimensions are listed in Table 3. We design the following walk reward functions to make the 700-actuator full-body model to walk forward based on a reference walking trajectory from motion capture data:

$$r_{\text{walk}} = 50 \cdot r_{\text{qpos}} + 0.1 \cdot r_{\text{qvel}} + 50 \cdot r_{\text{act}} + 5 \cdot r_{\text{vel}} + 100 \cdot r_{\text{healthy}}, \tag{32}$$

where $r_{\text{qpos}}$ penalizes the squared error of between model and reference joint position; $r_{\text{qvel}}$ penalizes the squared error of between model and reference joint velocity; $r_{\text{act}}$ penalizes the $l_2$-norm of the total actuator forces; $r_{\text{vel}}$ penalizes the squared error of between model and reference center-of-mass velocity; $r_{\text{healthy}}$ encourages the model not to fall down and deviate from the reference trajectory.

---

[10] https://github.com/facebookresearch/humenv
[11] https://github.com/carlosferrazza/humanoid-bench
[12] https://github.com/MyoHub/myosuite
[13] https://github.com/vittorione94/ostrichrl

Table 3: States in the MS-Human-700-Walking environments.

| State | Dimension |
|---|---|
| Joint position | 85 |
| Joint velocity | 85 |
| Joint acceleration | 85 |
| Actuator activation | 700 |
| Actuator force | 700 |
| Actuator length | 700 |
| Actuator velocity | 700 |
| Simulation time | 1 |
| Phase in walking period | 1 |
| Pelvis position | 3 |
| Sternum position | 3 |
| Joint position error | 85 |

**MS-Human-700-Run.** We design the following walk reward functions to make the 700-actuator full-body model to run forward based on a reference trajectory from CMU Graphics Lab Motion Capture Database[14] (Subject #2, Trial #3):

$$r_{\text{run}} = 10 \cdot r_{\text{qpos}} + 100 \cdot r_{\text{healthy}}, \tag{33}$$

with reward terms defined same as `MS-Human-700-Walk` under different reference trajectory.

**MS-Human-700-Dance.** We design the following walk reward functions to make the 700-actuator full-body model to perform ballet dancing based on a clip of reference trajectory from CMU Graphics Lab Motion Capture Database (Subject #5, Trial #9):

$$r_{\text{dance}} = 5 \cdot r_{\text{qpos}} + 100 \cdot r_{\text{xpos}} + 100 \cdot r_{\text{healthy}}, \tag{34}$$

where $r_{\text{xpos}}$ penalizes the squared error of between model and reference body position. The remaining reward terms are defined same as `MS-Human-700-Walk` under different reference trajectory.

## C  ADDITIONAL EXPERIMENTS

**Comparison with PPO.** We follow the official HumanoidBench repository and evaluate PPO on the Unitree H1 Run/Balance task using Stable-Baselines3. We observe that PPO exhibits limited reward improvement, which is consistent with the findings reported in the HumanoidBench paper. QFLEX substantially outperforms this widely used baseline.

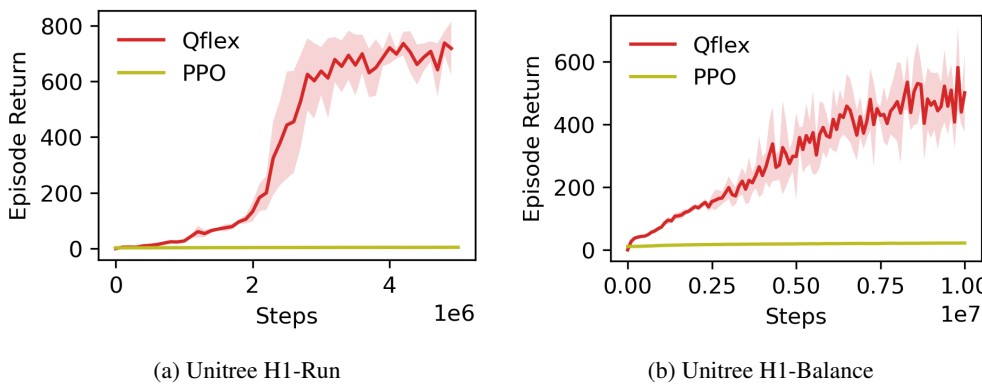

(a) Unitree H1-Run  (b) Unitree H1-Balance

Figure 6: Learning curve of of QFLEX and PPO on Unitree H1 tasks

**Comparison with flow-based online RL.** We compare QFLEX with FlowRL (Lv et al., 2025) on the Unitree H1–Balance task, which is also evaluated in the FlowRL paper. Since the official FlowRL

---

[14]https://mocap.cs.cmu.edu/

| Method | MyoLeg-Walk (80 actuators) | MS-Human-700-Walk (700 actuators) |
|--------|---------------------------|-----------------------------------|
| QFLEX  | **34.26 ± 2.84**          | **307.51 ± 20.37**                |
| CrossQ | 38.47 ± 3.49              | 356.87 ± 22.57                    |

Table 4: Energy efficiency measured by total actuator activation (lower is better). Results shows mean performances with one standard deviation.

implementation[15] supports only single-environment training, we run all algorithms in a single environment and align network architectures and training hyperparameters. We successfully reproduce the FlowRL performance reported in the original paper, and QFLEX substantially outperforms this baseline, highlighting its systematic advantages over FlowRL in high-dimensional continuous control (see Figure 7).

**Comparison with intrinsic motivation-based RL.** We additionally compare QFLEX against MaxInfoRL (Sukhija et al., 2024), an intrinsic-motivation method that promotes exploration via estimated information gain and includes evaluations on Unitree-H1 robots. We refer to the official implementation[16] and use the MaxInfoSAC variant, which is the primary version evaluated in the original paper. We observe QFLEX significantly outperforms MaxInfoRL on the Unitree H1–Balance task (see Figure 7). We consider intrinsic motivation-based RL methods encourage exploration by modifying the learning objective, but they do not directly address the challenge of inefficient sampling in high-dimensional continuous control.

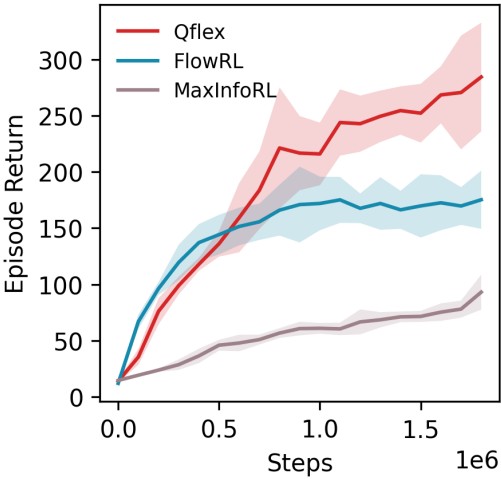

Figure 7: Learning curve of algorithms on Unitree H1-Balance task. Algorithms are trained on single environment. Results show mean performances with one standard deviation of 5 independent runs.

**Ablation over exploration strategy.** On the MS-Human-700–Walk task, we directly ablate the flow-based exploration strategy against a Gaussian-based alternative. QFLEX significantly outperforms the Gaussian-exploration variant, demonstrating the systematic advantage of flow-based exploration in high-dimensional continuous control (see Figure 8).

**Energy efficiency.** In the MyoLeg–Walk and MS-Human-700–Walk tasks, we compare the total muscle activation of QFLEX with CrossQ (the strongest Gaussian-based policy). QFLEX achieves substantially lower total muscle activation, demonstrating the superior energy efficiency enabled by flow-based exploration (see Table 4).

**Runtime analysis.** On the MyoLeg–Walk task, we compare the runtime of QFLEX with Gaussian-based and diffusion-based baselines on an NVIDIA GeForce RTX 4090 D GPU. QFLEX achieves

---

[15] https://github.com/bytedance/FlowRL
[16] https://github.com/sukhijab/maxinforl_jax

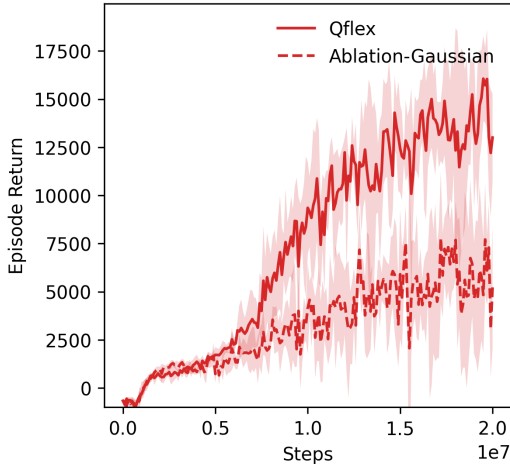

Figure 8: Ablation over exploration strategy on MS-Human-700-Walk task. Results show mean performances with one standard deviation of 5 independent runs.

| Method | Training time | Per-step deployment time |
|--------|---------------|--------------------------|
| QFLEX | $52.94 \pm 0.29$min | $0.49 \pm 0.19$ms |
| SDAC | $82.26 \pm 0.71$min | $4.07 \pm 0.58$ms |
| DACER | $75.06 \pm 0.08$min | $0.54 \pm 0.28$ms |
| QSM | $48.36 \pm 0.24$min | $2.95 \pm 0.39$ms |
| SAC | $29.4 \pm 0.12$min | $0.11 \pm 0.05$ms |
| CrossQ | $34.83 \pm 0.04$min | $0.13 \pm 0.05$ms |

Table 5: Runtime comparison of algorithms on MyoLeg-Walk task. Results shows mean performances with one standard error.

comparable or lower runtime relative to all evaluated diffusion-based methods (see Table 5,). Although its runtime is higher than that of Gaussian-based baselines, this overhead is acceptable given the substantial performance gains and remains well within real-time control requirements.

**Correlation of** QFLEX **exploration.** We conduct the exploration analysis on the MyoLeg–Walk task. Because the flow-based distribution is difficult to visualize directly, we approximate QFLEX's exploration noise by computing the standard deviation of 1,000 sampled actions at each timestep. QFLEX exhibits structured correlations across action dimensions, in contrast to isotropic Gaussian noise (see Figure 9). We further observe strong correlations among actuators within the same anatomical groups, for example, the gluteus maximus (glmax1, glmax2, glmax3) and the gastrocnemius (gaslat, gasmed). These patterns provide evidence that QFLEX performs directed exploration informed by both task structure and system morphology.

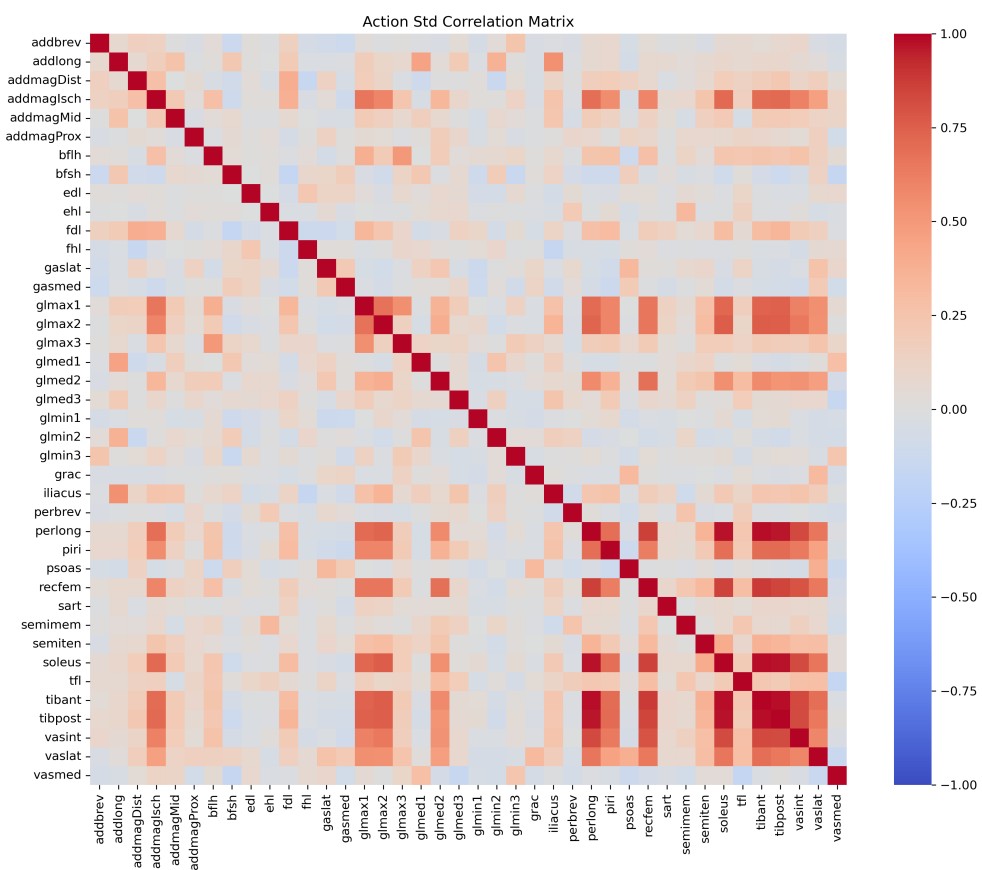

Figure 9: Correlation matrix of QFLEX exploration over right lowerbody muscles in MyoLeg-Walk.

