# OpenReview forum: "Scalable Exploration for High-Dimensional Continuous Control via Value-Guided Flow"
_ICLR.cc/2026/Conference — ICLR 2026 Poster_

### Official Review · Reviewer_cSUp · 2025-10-17

**Soundness:** 3
**Presentation:** 3
**Contribution:** 3
**Rating:** 6
**Confidence:** 4

**Summary:**

This paper introduces QFLEX, a novel reinforcement learning method that uses a Q-guided probability flow for directed exploration in high-dimensional action spaces. The approach is motivated by the well-justified problem of vanishing exploration with undirected noise in over-actuated systems. The ideas are clearly presented, and the experimental evaluation is comprehensive, spanning from typical humanoids like Unitree H1 to a challenging 700-actuator musculoskeletal model.

**Strengths:**

1. The core idea of using a learned probability flow for value-directed exploration is novel and well-motivated. It effectively addresses a key limitation (vanishing signal) of isotropic exploration in high-dimensional spaces.
2. The paper is generally well-written. The main ideas are clear and easy to follow.
3. The empirical validation is a major strength. The benchmarks are diverse and highly relevant, and the successful application to the full-body musculoskeletal model (MS-Human-700) provides strong evidence for the method's scalability and practical potential.

**Weaknesses:**

1. The method incurs a non-trivial computational cost. Obtaining an exploratory action $a^{(1)}$ requires calculating the gradient of $Q$ with respect to $a^{(\frac{n-1}{N})}$ for N times (Eq. 14).
Quantifying the wall-clock time compared to baselines would strengthen this analysis.
2. The comparison focuses on Gaussian-based and diffusion-based RL baselines. Including a comparison with other advanced exploration strategies for high-dimensional spaces (e.g., those based on intrinsic motivation or curiosity) could better situate QFLEX within the broader RL exploration literature and highlight its specific advantages.
3. The intuition behind learning a flow model, as opposed to simply taking a fixed number of gradient ascent steps from the current policy $\pi^{(0)}$, could be more thoroughly justified. An ablation study directly comparing the learned flow against a simple multi-step gradient ascent baseline would help isolate the benefit of the flow matching component.

**Questions:**

1. For the case analysis of vanishing exploration in high DoF settings, the result seems to depend on the choice of the link length $L/|\mathcal{A}|$. If we choose the length as $L/\sqrt{|\mathcal{A}|}$, we can not get the conclusion. Could the authors comment on the choice of $l_i=L/|\mathcal{A}|$.
2. The algorithm simultaneously trains the Q-function, the initial Gaussian policy $\pi^{(0)}$, and the flow model $v_w$. Since both the source and target of this flow are non-stationary, could the author provide more insight or empirical evidence on why this joint training is stable?

---

> ### Author Response · Authors · 2025-11-22
> **Rebuttal (1/2)**
>
> Thanks for your appreciations of our work! Below we clarify your concerns. Please refer to the revised pdf to see our additional experimental results.
>
> > **W1:** The method incurs a non-trivial computational cost. Obtaining an exploratory action $\boldsymbol{a}^{(1)}$ requires calculating the gradient of $Q$ with respect to $\boldsymbol{a}^{(\frac{n-1}{N})}$ for $N$ times (Eq. 14). Quantifying the wall-clock time compared to baselines would strengthen this analysis.
> >
> > **W3:** The intuition behind learning a flow model, as opposed to simply taking a fixed number of gradient ascent steps from the current policy $\pi^{(0)}$, could be more thoroughly justified. An ablation study directly comparing the learned flow against a simple multi-step gradient ascent baseline would help isolate the benefit of the flow matching component.
>
> **Reply to W1 & W3**
>
> We adopt a learned flow model rather than directly applying gradient ascent primarily to improve deployment efficiency, as taking multiple gradient steps is more expensive than feedforward passes. Flow matching effectively decouples the construction of the Q-guided flow during training from the ODE-solving process at deployment. This allows us to use a sufficiently large number of gradient steps when constructing the Q-guided flow for policy learning, while maintaining fast deployment by adjusting the number of integration steps in the ODE solver.
>
> On the MyoLeg–Walk task, we compare the runtime of Qflex with Gaussian-based and diffusion-based baselines on an NVIDIA GeForce RTX 4090 D GPU. Qflex achieves comparable or lower runtime relative to all evaluated diffusion-based methods (see the table below or Table 5 in Appendix C). Although its runtime is higher than that of Gaussian-based baselines, this overhead is acceptable given the substantial performance gains and remains well within real-time control requirements.
>
> | Method | Training time       | Per-step deployment time |
> |--------|---------------------|--------------------------|
> | Qflex  | $52.94 \pm 0.29$min | $0.49\pm0.19$ms          |
> | SDAC   | $82.26 \pm 0.71$min | $4.07\pm 0.58$ms         |
> | DACER  | $75.06 \pm 0.08$min | $0.54\pm0.28$ms          |
> | QSM    | $48.36 \pm 0.24$min | $2.95\pm0.39$ms          |
> | SAC    | $29.4 \pm 0.12$min  | $0.11\pm0.05$ms          |
> | CrossQ | $34.83 \pm 0.04$min | $0.13\pm0.05$ms          |
>
> > **W2:** The comparison focuses on Gaussian-based and diffusion-based RL baselines. Including a comparison with other advanced exploration strategies for high-dimensional spaces (e.g., those based on intrinsic motivation or curiosity) could better situate QFLEX within the broader RL exploration literature and highlight its specific advantages.
>
> **Reply to W2**
>
> We additionally compare Qflex against MaxInfoRL [1], an intrinsic-motivation method that promotes exploration via estimated information gain and includes evaluations on Unitree-H1 robots. We use the MaxInfoSAC variant, which is the primary version evaluated in the original paper. We observe Qflex significantly outperforms MaxInfoRL on the Unitree H1–Balance task (see Figure 7 in Appendix C). We consider intrinsic motivation-based RL methods encourage exploration by modifying the learning objective, but they do not directly address the challenge of inefficient sampling in high-dimensional continuous control.
>
> [1] Sukhija, Bhavya, et al. "MaxInfoRL: Boosting exploration in reinforcement learning through information gain maximization." The Thirteenth International Conference on Learning Representations.

---

> > ### Author Response · Authors · 2025-11-22
> > **Rebuttal (2/2)**
> >
> > > **Q1:** For the case analysis of vanishing exploration in high DoF settings, the result seems to depend on the choice of the link length $L/|\mathcal{A}|$. If we choose the length as $L/\sqrt{|\mathcal{A}|}$
> > , we can not get the conclusion. Could the authors comment on the choice of $l_i=L/|\mathcal{A}|$.
> >
> > **Reply to Q1**
> >
> > We set each link length to $L/|\mathcal{A}|$ to ensure fair comparison across different action dimensionalities, so that all linkages have the same total length and differ only in the number of joints.
> >
> > > **Q2:** The algorithm simultaneously trains the Q-function, the initial Gaussian policy $\pi^{(0)}$, and the flow model $\boldsymbol{v}_w$. Since both the source and target of this flow are non-stationary, could the author provide more insight or empirical evidence on why this joint training is stable?
> >
> > **Reply to Q2**
> >
> > We consider following the Q-function gradient is an effective mechanism to produce informative samples for Q-function learning. When the gradient is unreliable, action samples helps the Q-network to correct potential overestimation using the reward feedback. This process progressively refines the Q-function until the gradient becomes reliable, where following this direction we can sample actions with high control performances.
> >
> >
> > Our algorithm analysis demonstrates that Qflex obtain this directed exploration capability during early training stages. Across all benchmarks, Qflex starts to produce actions with higher Q value than Gaussian sampling almost at the beginning of the training procedure (see Figure 4 in the main paper).
> >
> > Our comprehensive experimental results also demonstrate stable training. Using the same hyperparameters across all benchmarks, Qflex consistently outperforms the baseline methods (see Figures 2 and 3(a) in the main paper).

---

> ### Comment · Reviewer_cSUp · 2025-11-25
>
> Thank the authors for the revision. The reply addressed my concerns and I would like to raise my score to 8.

---

> > ### Author Response · Authors · 2025-11-26
> >
> > Thank you for raising your score! We will continue to improve our work by incorporating the new results.

---

### Official Review · Reviewer_987M · 2025-10-29

**Soundness:** 3
**Presentation:** 3
**Contribution:** 3
**Rating:** 4
**Confidence:** 4

**Summary:**

The paper introduces Qflex, a reinforcement learning algorithm designed for efficient exploration in high-dimensional continuous control. Exploration in such systems is challenging because Gaussian noise fails to explore efficiently, particularly in over-actuated settings (e.g. musculoskeletal control). Qflex addresses this by aligning exploration with value-function gradients through a probability flow formulation. The method directs exploration toward high-value regions by integrating a value-guided velocity field. Experiments on several high-dimensional benchmarks demonstrate improved sample efficiency and control performance over Gaussian and other exploration baselines.

**Strengths:**

- The idea of aligning exploration with the value function through a flow-based transformation is well grounded and connects flow matching with reinforcement learning in a principled way.
- The method achieves state-of-the-art performance on several high-dimensional control benchmarks, including tasks with >700 actuators.
- Qflex integrates easily into standard actor-critic frameworks such as SAC, without architectural modifications.

**Weaknesses:**

- While applying flow matching to online RL for exploration is interesting, closely related ideas appear in [1] and [2]. The main novelty lies in scaling to high-dimensional systems rather than a fundamentally new formulation.
- The paper does not analyze how value-guided flows improve exploration or in which regimes they outperform Gaussian or latent-space baselines.
- The qualitative figure contrasting Gaussian vs. Qflex exploration is illustrative but lacks quantitative evaluation or a controlled ablation demonstrating the behavioral difference.
- Metrics such as energy efficiency would help interpret whether exploration leads to more structured or efficient control.

[1]: McAllister, David, et al. "Flow matching policy gradients." *arXiv preprint arXiv:2507.21053* (2025).

[2]: Lv, Lei, et al. "Flow-Based Policy for Online Reinforcement Learning." *arXiv preprint arXiv:2506.12811* (2025).

**Questions:**

1. How does Qflex differ theoretically and empirically from [1] and [2]? Are the reported improvements due to the use of value-guided flows? A side-by-side comparison on shared benchmarks would clarify novelty.
2. Can the authors provide a small-scale toy controlled example (e.g., a 50-100 DOF) comparing Gaussian vs. Qflex noise? Visualizing trajectories and action-space evolution would concretely illustrate how Qflex produces more directed exploration. Additionally, can the authors discuss how much Qflex depends on the value function at the beginning of the training (as the critic is inaccurate) and when it transitions from random (Gaussian-like) to value-guided? How far into the future (short vs. long-term value) does the velocity field effectively look?
3. Does Qflex noise exhibit the same correlation structure as the learned action covariance? Is it aligned with principal action subspaces or dependent on state geometry? This would help understanding whether Qflex adaptively focuses exploration on effective actuator subsets.
4. How does solving the flow ODE scale with action dimension |A|? Does QFlex incur additional runtime or convergence cost compared to SAC, or are gains due to more efficient sampling?
5. In over-actuated systems, exploration efficiency is often correlated with energy-efficient policies (as shown in [3]). Does Qflex yield more energy-efficient or smoother action trajectories?
6. Are all baselines trained under identical regimes (fully online)? How are comparisons to DEP-RL (which is an offline method) made? Would Qflex still outperform other exploration baseline if trained with PPO instead of SAC?

[1]: McAllister, David, et al. "Flow matching policy gradients." *arXiv preprint arXiv:2507.21053* (2025).

[2]: Lv, Lei, et al. "Flow-Based Policy for Online Reinforcement Learning." *arXiv preprint arXiv:2506.12811* (2025).

[3]: Chiappa, Alberto Silvio, et al. "Latent exploration for reinforcement learning." *Advances in Neural Information Processing Systems* 36 (2023): 56508-56530.

---

> ### Author Response · Authors · 2025-11-22
> **Rebuttal (1/3)**
>
> Thanks for your insightful feedback. Below we clarify your concerns. Please refer to the revised pdf to see our additional experimental results.
>
> > **W1:** While applying flow matching to online RL for exploration is interesting, closely related ideas appear in [1] and [2]. The main novelty lies in scaling to high-dimensional systems rather than a fundamentally new formulation.
> >
> > **Q1:** How does Qflex differ theoretically and empirically from [1] and [2]? Are the reported improvements due to the use of value-guided flows? A side-by-side comparison on shared benchmarks would clarify novelty.
> >
> > [1] McAllister, David, et al. "Flow matching policy gradients." arXiv preprint arXiv:2507.21053 (2025).
> >
> > [2] Lv, Lei, et al. "Flow-Based Policy for Online Reinforcement Learning." arXiv preprint arXiv:2506.12811 (2025).
>
> **Reply to W1 & Q1**
>
> We argue that QFlex’s strong performance in high-dimensional continuous control stems from its fundamentally different formulation compared to prior diffusion- and flow-based online RL methods, as discussed at the end of Section 5 (lines 319–323). Qflex is related to FPO[1] and FlowRL[2] in the use of flow matching, but differs substantially in the construction of both source and target distribution.
>
> *Source distribution*
>
> FPO and FlowRL adopt the standard Gaussian source distribution common in diffusion/flow-based RL, which may lie far from the target policy distribution and provide limited guidance for training the velocity field. In contrast, Qflex employs a learnable source distribution, yielding more informative initializations for constructing the Q-guided flow and simplifying the velocity-field learning.
>
> *Target distribution*
>
> FPO and FlowRL both learn a weighted distribution of actions sampled by the policy, where the performance of the target distribution is upper-bounded by the current policy. Qflex instead constructs the target distribution directly by following the Q-guided flow, which yields an explicit policy-improvement guarantee (see Proposition 1).
>
> *Performance comparison*
>
> We follow the official repositories to run both FPO and FlowRL under the same experimental settings. However, FPO currently does not support high-dimensional tasks such as humanoid control due to its use of an exp-square operation in the log-likelihood ratio computation, which is an [unresolved issue](https://github.com/akanazawa/fpo/issues/4) acknowledged by the authors.
>
> We compare Qflex with FlowRL on the Unitree H1–Balance task, which is also evaluated in the FlowRL paper. Since the official FlowRL implementation supports only single-environment training, we run all algorithms in a single environment and align network architectures and training hyperparameters. We successfully reproduce the FlowRL performance reported in the original paper, and Qflex substantially outperforms this baseline, highlighting its systematic advantages over FlowRL in high-dimensional continuous control (see Figure 7 in Appendix C).
>
> > **W2:** The paper does not analyze how value-guided flows improve exploration or in which regimes they outperform Gaussian or latent-space baselines.
>
> **Reply to W2**
>
> Our theoretical analysis and empirical results demonstrate that sampling from value-guided flows consistently enhances exploration.
>
> 1. Theoretically, we show that sampling along the value-guided flows leads to better policy improvement compared against Gaussian-based policy (see Proposition 1).
> 2. Our algorithm analysis demonstrates that sampling over value-guided flows consistently yields better action samples during whole training procedure across all our evaluated benchmarks (see Figure 4 in the main paper).
> 3. We show that Qflex significantly outperforms both Gaussian and latent-space exploration strategies across a wide range of high-dimensional continuous-control benchmarks, demonstrating consistent advantages (see Figures 2 and 3(a) in the main paper).
>
> > **W3:** The qualitative figure contrasting Gaussian vs. Qflex exploration is illustrative but lacks quantitative evaluation or a controlled ablation demonstrating the behavioral difference.
>
> **Reply to W3**
>
> On the MS-Human-700–Walk task, we directly ablate the flow-based exploration strategy against a Gaussian-based alternative. Qflex significantly outperforms the Gaussian-exploration variant, demonstrating the systematic advantage of flow-based exploration in high-dimensional continuous control (see Figure 8 in Appendix C).

---

> ### Author Response · Authors · 2025-11-22
> **Rebuttal (2/3)**
>
> > **W4:** Metrics such as energy efficiency would help interpret whether exploration leads to more structured or efficient control.
> >
> > **Q6:** In over-actuated systems, exploration efficiency is often correlated with energy-efficient policies (as shown in [3]). Does Qflex yield more energy-efficient or smoother action trajectories?
> >
> > [3] Chiappa, Alberto Silvio, et al. "Latent exploration for reinforcement learning." Advances in Neural Information Processing Systems 36 (2023): 56508-56530.
>
> **Reply to W4 & Q6**
>
> In the MyoLeg–Walk and MS-Human-700–Walk tasks, we compare the total muscle activation (lower is better) of Qflex with CrossQ (the strongest Gaussian-based policy). Qflex achieves substantially lower total muscle activation, demonstrating the superior energy efficiency enabled by flow-based exploration (see the table below or Table 4 in Appendix C).
>
> | Method | MyoLeg-Walk (80 actuators)    | MS-Human-700-Walk (700 actuators) |
> |--------|-------------------------------|-----------------------------------|
> | Qflex  | $\boldsymbol{34.26 \pm 2.84}$ | $\boldsymbol{307.51 \pm 20.37}$   |
> | CrossQ | $38.47 \pm 3.49$              | $356.87 \pm 22.57$                |
>
> > **Q2:** Can the authors provide a small-scale toy controlled example (e.g., a 50-100 DOF) comparing Gaussian vs. Qflex noise? Visualizing trajectories and action-space evolution would concretely illustrate how Qflex produces more directed exploration.
>
> **Reply to Q2**
>
> In Figure 1, we visualize the exploration distributions of Gaussian (green) and Qflex (red) under varying action dimensionalities. We observe that Qflex consistently produces informative, high-value samples in high-dimensional action spaces, whereas Gaussian-based exploration fails.
>
> > **Q3:** Additionally, can the authors discuss how much Qflex depends on the value function at the beginning of the training (as the critic is inaccurate) and when it transitions from random (Gaussian-like) to value-guided? How far into the future (short vs. long-term value) does the velocity field effectively look?
>
> **Reply to Q3**
>
> *Exploration under inaccurate value*
>
> We consider following the Q-function gradient is an effective mechanism to produce informative samples for Q-function learning. When the gradient is unreliable, action samples helps the Q-network to correct potential overestimation using the reward feedback. This process progressively refines the Q-function until the gradient becomes reliable, where following this direction we can sample actions with high control performances.
>
> *When Qflex transit to value-aligned*
>
> Our algorithm analysis also demonstrates that Qflex obtain this directed exploration capability during early training stages. Across all benchmarks, Qflex starts to produce actions with higher Q value than Gaussian sampling almost at the beginning of the training procedure (see Figure 4 in the main paper).
>
> *Value function type*
>
> Our method is derived under the infinite-horizon MDP setting, where the value function is defined as an infinite sum of discounted rewards. In all experiments, we set the discount factor of the learned Q-function to 0.99.
>
> > **Q4:** Does Qflex noise exhibit the same correlation structure as the learned action covariance? Is it aligned with principal action subspaces or dependent on state geometry? This would help understanding whether Qflex adaptively focuses exploration on effective actuator subsets.
>
> **Reply to Q4**
>
> We conduct the exploration analysis on the MyoLeg–Walk task. Because the flow-based distribution is difficult to visualize directly, we approximate Qflex’s exploration noise by computing the standard deviation of 1,000 sampled actions at each timestep. Qflex exhibits structured correlations across action dimensions, in contrast to isotropic Gaussian noise (see Figure 9 in Appendix C). We further observe strong correlations among actuators within the same anatomical groups, for example, the gluteus maximus (glmax1, glmax2, glmax3) and the gastrocnemius (gaslat, gasmed). These patterns provide evidence that Qflex performs directed exploration informed by both task structure and system morphology.

---

> ### Author Response · Authors · 2025-11-22
> **Rebuttal (3/3)**
>
> > **Q5:** How does solving the flow ODE scale with action dimension |A|? Does Qflex incur additional runtime or convergence cost compared to SAC, or are gains due to more efficient sampling?
>
>
> **Reply to Q5**
>
> We consider the ODE solving time of Qflex to be moderate for flow-based methods, which are widely used in computer vision. Our highest-dimensional task ($|\mathcal{A}|=700$) is still smaller than a $28\times28$ grayscale image.
>
> On the MyoLeg–Walk task, we compare the runtime of Qflex with Gaussian-based and diffusion-based baselines on an NVIDIA GeForce RTX 4090 D GPU. Qflex achieves comparable or lower runtime relative to all evaluated diffusion-based methods (see the table below or Table 5 in Appendix C). Although its runtime is higher than that of Gaussian-based baselines, this overhead is acceptable given the substantial performance gains and remains well within real-time control requirements.
>
> | Method | Training time       | Per-step deployment time |
> |--------|---------------------|--------------------------|
> | Qflex  | $52.94 \pm 0.29$min | $0.49\pm0.19$ms          |
> | SDAC   | $82.26 \pm 0.71$min | $4.07\pm 0.58$ms         |
> | DACER  | $75.06 \pm 0.08$min | $0.54\pm0.28$ms          |
> | QSM    | $48.36 \pm 0.24$min | $2.95\pm0.39$ms          |
> | SAC    | $29.4 \pm 0.12$min  | $0.11\pm0.05$ms          |
> | CrossQ | $34.83 \pm 0.04$min | $0.13\pm0.05$ms          |
>
> > **Q7:** Are all baselines trained under identical regimes (fully online)? How are comparisons to DEP-RL (which is an offline method) made?
>
> We align all common components across baselines, including network size, learning rate, gradient steps, number of environments, and discount factor. We respectfully disagree with the characterization of DEP-RL [4] as an offline RL method. As described in the original paper, DEP-RL is an online RL approach that interleaves policy learning with DEP-based exploration, making it directly comparable to Qflex in the same manner as other online RL baselines.
>
> [4] Schumacher, Pierre, et al. "DEP-RL: Embodied Exploration for Reinforcement Learning in Overactuated and Musculoskeletal Systems." The Eleventh International Conference on Learning Representations.
>
> > **Q8:** Would Qflex still outperform other exploration baseline if trained with PPO instead of SAC?
>
> We clarify that our implementation of Qflex is not built on SAC or PPO. Instead, Algorithm 1 employs a minimal actor–critic framework without entropy or KL regularization. This design choice allows us to clearly isolate and demonstrate the benefits of flow-based exploration.

---

> > ### Comment · Reviewer_987M · 2025-11-22
> >
> > I would like to thank the authors for their detailed responses. Most of my concerns have been addressed, and the new experiments further support the value of this work. I have therefore decided to raise my score from 4 to 8.

---

> > > ### Author Response · Authors · 2025-11-22
> > >
> > > Thank you for raising your score! We will continue to improve the paper based on your suggestions.

---

### Official Review · Reviewer_i8Yj · 2025-10-30

**Soundness:** 3
**Presentation:** 3
**Contribution:** 3
**Rating:** 8
**Confidence:** 3

**Summary:**

The work addresses a critical challenge in high-dimensional continuous control: achieving efficient exploration in complex state–action spaces. Common practices such as adding Gaussian noise to policy actions often fail to promote meaningful exploration—an issue well recognized in the community and clearly demonstrated by the authors. In contrast, the proposed method leverages action samples guided by estimated Q-value improvements, enabling more directed and effective exploration.

The method outperforms Gaussian-based and diffusion-based RL baselines across several high-dimensional locomotion tasks and even achieves control of a 700-actuator full-body human musculoskeletal model, which is particularly impressive and highlights the practical advantages of the proposed approach.

**Strengths:**

The idea is well-motivated, clearly established, and novel. The problem it addresses is central to scaling reinforcement learning policies to more complex and realistic control tasks. Leveraging Q-guided flows to generate actions for directed exploration is both natural and insightful, with its benefits demonstrated across several high-dimensional tasks that are typically challenging for RL policies to learn from scratch. Moreover, the method introduces only minimal modifications to a standard actor–critic framework, further enhancing its practicality and applicability.

**Weaknesses:**

The primary weakness of the proposed method lies in its strong reliance on accurate Q-gradient estimation. During early training stages, or in scenarios where Q-gradients are unreliable, the resulting action updates may misguide exploration and ultimately hinder policy improvement. Although the paper discusses the use of batch normalization to stabilize Q-learning, a more thorough analysis of potential failure cases or mitigation strategies would further strengthen the work.

**Questions:**

The experiments are primarily conducted on tasks with dense reward signals, which are common in locomotion benchmarks. It might be unclear how the method would perform in sparse-reward environments, where accurate Q-function learning is more challenging. In such settings, would Q-guided flows still yield effective exploration, or would their reliance on precise Q-gradients diminish their advantages?

From Algorithm 2, the policy appears to always execute the final transported action from the Q-guided flow when interacting with the environment. This design seems heavily biased toward exploitation, as each exploratory step moves strictly in the direction of higher estimated value. Could this cause the exploration to collapse prematurely and converge to sub-optimal policies, especially when the Q-gradient is inaccurate early in training? Have you considered sampling actions from intermediate states along the flow trajectory—rather than only at the terminal point—to preserve exploration breadth? If so, what empirical effects and trade-offs did you observe?

---

> ### Author Response · Authors · 2025-11-22
> **Rebuttal (1/1)**
>
> Thanks for your appreciations of our work! Below we clarify your concerns. Please refer to the revised pdf to see our additional experimental results.
>
> > **W1:** The primary weakness of the proposed method lies in its strong reliance on accurate Q-gradient estimation. During early training stages, or in scenarios where Q-gradients are unreliable, the resulting action updates may misguide exploration and ultimately hinder policy improvement. Although the paper discusses the use of batch normalization to stabilize Q-learning, a more thorough analysis of potential failure cases or mitigation strategies would further strengthen the work.
> >
> > **Q2:** From Algorithm 2, the policy appears to always execute the final transported action from the Q-guided flow when interacting with the environment. This design seems heavily biased toward exploitation, as each exploratory step moves strictly in the direction of higher estimated value. Could this cause the exploration to collapse prematurely and converge to sub-optimal policies, especially when the Q-gradient is inaccurate early in training?
>
> **Reply to W1 & Q2**
>
> We consider following the Q-function gradient is an effective mechanism to produce informative samples for Q-function learning. When the gradient is unreliable, action samples helps the Q-network to correct potential overestimation using the reward feedback. This process progressively refines the Q-function until the gradient becomes reliable, where following this direction we can sample actions with high control performances.
>
> Our algorithm analysis also demonstrates that Qflex obtain this directed exploration capability during early training stages. Across all benchmarks, Qflex starts to produce actions with higher Q value than Gaussian sampling almost at the beginning of the training procedure (see Figure 4 in the main paper).
>
> > **Q1:** The experiments are primarily conducted on tasks with dense reward signals, which are common in locomotion benchmarks. It might be unclear how the method would perform in sparse-reward environments, where accurate Q-function learning is more challenging. In such settings, would Q-guided flows still yield effective exploration, or would their reliance on precise Q-gradients diminish their advantages?
>
> **Reply to Q1**
>
> Beside imitation-based locomotion, our main paper also includes benchmarks with sparse rewards. SMPL Humanoid–Jump task adopts a sparse-reward formulation in its [official implementation](https://github.com/facebookresearch/humenv/blob/0548761a19b783c6a3548a14a8c4ea1ca35e94af/humenv/rewards.py#L12):
>
> $r=r_{\text{jump}} \times r_{\text{upright}} \times r_{\text{velocity}},$
>
> where $r_{\text{jump}}$ and $r_{\text{velocity}}$ take non-negligible values only when the humanoid’s jump height or upward velocity exceeds a specified threshold. Most baselines struggle to achieve reward improvement under this sparse setting, whereas Qflex explores efficiently and significantly outperforms all baselines (see Figure 2 in the main paper).
>
> > **Q3:** Have you considered sampling actions from intermediate states along the flow trajectory—rather than only at the terminal point—to preserve exploration breadth? If so, what empirical effects and trade-offs did you observe?
>
> **Reply to Q3:**
>
> Flow matching enforces consistency between the target distribution and the learned flow terminal point, but the intermediate states along the probability path do not necessarily correspond to the true Q-gradient trajectory. For this reason, we sample actions only from the terminal point of the flow. The trade-off between directed and undirected exploration is determined by the number of finite gradient steps: different step counts correspond to sampling from different intermediate states along the Q-guided probability flow. Our ablation over this hyperparameter shows that Qflex maintains broadly comparable learning performance across a reasonable range of gradient-step choices (see Figure 5(a) in the main paper).

---

> ### Comment · Reviewer_i8Yj · 2025-11-24
>
> Thank you for the response. While some of my questions remain only partially addressed, I agree that within the specific application domain, the authors provide sufficient evidence to demonstrate the method’s effectiveness. I do not wish for my nitpicking to overshadow the value of the work, so I will maintain my original score and confidence level.

---

> > ### Author Response · Authors · 2025-11-26
> >
> > Thank you for the feedback! We will continue to improve the paper based on your suggestions.

---

### Official Review · Reviewer_s5BZ · 2025-11-01

**Soundness:** 3
**Presentation:** 4
**Contribution:** 3
**Rating:** 6
**Confidence:** 3

**Summary:**

This paper proposes a scalable online reinforcement learning method （QFLEX）to efficiently explore high-dimensional continuous control systems. Through a case analysis, the paper demonstrates that the undirected stochasticity leads to vanishing exploration in high-dimensional continuous control. The proposed method, QFLEX, outperforms six other baselines across multiple high-dimensional control benchmarks. Additionally, QFLEX successfully controls a full-body musculoskeletal model with 700 actuators to perform complex tasks, such as walking and ballet dancing.

**Strengths:**

1.	The paper presents clear and concise expression.
2.	The paper presents thorough comparative experiments, testing QFLEX on six benchmark tasks and comparing it with eight baseline methods.
3.	The proposed QFLEX method efficiently addresses exploration in high-dimensional action spaces, overcoming the inefficiencies of undirected exploration.

**Weaknesses:**

1. Although QFLEX performs well in simulated tasks, the paper does not provide sufficient validation on real-world systems.

**Questions:**

In the Unitree H1–Run/Balance task, the results presented in the paper indicate that QFLEX performs notably better than the other methods. Whether Proximal Policy Optimization (PPO) was tested as well ? It may strengthen the paper's contributions to include a comparison with PPO.

---

> ### Author Response · Authors · 2025-11-22
> **Rebuttal (1/1)**
>
> Thanks for your appreciations of our work! Below we clarify your concerns. Please refer to the revised pdf to see our additional experimental results.
>
> > **W1:** Although QFLEX performs well in simulated tasks, the paper does not provide sufficient validation on real-world systems.
>
> **Reply to W1**
>
> We agree that real-world deployment is an important direction for Qflex. In our simulated experiment, Qflex consistently outperforms extensive baselines on the Unitree-H1 platform, and we plan to extend these efforts to additional real-world robotic systems.
>
> > **Q1:** In the Unitree H1–Run/Balance task, the results presented in the paper indicate that QFLEX performs notably better than the other methods. Whether Proximal Policy Optimization (PPO) was tested as well ? It may strengthen the paper's contributions to include a comparison with PPO.
>
> **Reply to Q1**
>
> We follow the official HumanoidBench repository and evaluate PPO on the Unitree H1 Run/Balance task using Stable-Baselines3. We observe that PPO exhibits limited reward improvement, which is consistent with the findings reported in the HumanoidBench paper. Qflex substantially outperforms this widely used baseline (see Figure 6 in Appendix C).

---

### Meta-Review · Area_Chair_tpEZ · 2026-01-06

**Summary:**

The paper introduces Qflex, a reinforcement learning method for scalable exploration in high-dimensional continuous control that uses value-guided flows rather than traditional isotropic noise. All four reviewers recognized the merit of this work, with initial scores of 8, 6, 4, and 8 (before the OpenReview bug required score reversion). The reviewers converged on appreciating the well-motivated core idea, the comprehensive experimental validation across diverse high-dimensional benchmarks, and the demonstration on a 700-actuator musculoskeletal model.
The primary concerns raised involved: (1) computational costs and runtime scalability compared to baselines (Reviewer cSUp), (2) theoretical novelty relative to concurrent flow-based RL methods and the need for deeper analysis of when value-guided flows outperform alternatives (Reviewer 987M), (3) reliability of exploration when Q-gradients are inaccurate during early training (Reviewer i8Yj), and (4) validation on real-world systems and comparison with PPO (Reviewer s5BZ). Reviewer 987M initially scored the paper as marginally below threshold (4) due to concerns about novelty and insufficient mechanistic analysis, while the other reviewers were more positive from the outset.

**Reviewer Concerns:**

The authors provided a thorough rebuttal with substantial additional experimental results. Key addressed concerns include: the authors clarified fundamental differences from FPO and FlowRL in terms of source distribution design and target distribution construction, providing direct performance comparisons showing Qflex's advantages; they added runtime analysis demonstrating acceptable computational overhead relative to diffusion-based methods; they included energy efficiency metrics showing superior muscle activation patterns; they provided correlation structure analysis of the exploration noise demonstrating task-informed directed exploration; and they added comparisons with MaxInfoRL and PPO as requested.
Three of four reviewers (987M, cSUp, i8Yj) explicitly responded to the rebuttal. Reviewers 987M and cSUp both raised their scores to 8, indicating that their concerns were substantially addressed. Reviewer i8Yj maintained the score of 8, noting that while some questions remained partially addressed, the evidence was sufficient to demonstrate the method's value. Reviewer s5BZ did not participate in post-rebuttal discussion but had raised specific concerns about real-world validation and PPO comparison, both of which the authors addressed in their response.

**Reviewer Scores:**

Based on the rebuttal responses and the discussion that occurred, the AC assess the expected final scores as follows: Reviewer i8Yj would maintain their score of 8, as they explicitly stated this in their post-rebuttal comment; Reviewer 987M would score 8, as they explicitly raised their score from 4 to 8 after the rebuttal; Reviewer cSUp would score 8, as they explicitly raised their score from 6 to 8 after the rebuttal; and Reviewer s5BZ would likely maintain or increase their score from 6, given that the authors comprehensively addressed their concerns regarding PPO comparison and provided additional experimental validation, though no explicit response was received.

---

### Decision · Program_Chairs · 2026-01-26

Accept (Poster)